# Bioenergetic and Proteomic Profiling of Immune Cells in Myalgic Encephalomyelitis/Chronic Fatigue Syndrome Patients: An Exploratory Study

**DOI:** 10.3390/biom11070961

**Published:** 2021-06-29

**Authors:** Paula Fernandez-Guerra, Ana C. Gonzalez-Ebsen, Susanne E. Boonen, Julie Courraud, Niels Gregersen, Jesper Mehlsen, Johan Palmfeldt, Rikke K. J. Olsen, Louise Schouborg Brinth

**Affiliations:** 1Research Unit for Molecular Medicine, Department of Clinical Medicine, Aarhus University and Aarhus University Hospital, 8200 Aarhus, Denmark; ac.gonzalez@clin.au.dk (A.C.G.-E.); nig@clin.au.dk (N.G.); johan.palmfeldt@clin.au.dk (J.P.); 2KMEB, Department of Endocrinology, Odense University Hospital and Department of Clinical Research, University of Southern Denmark, 5000 Odense, Denmark; 3Department of Clinical Genetics, Odense University Hospital, 5000 Odense, Denmark; susanne.eriksen.boonen@rsyd.dk; 4Section for Clinical Mass Spectrometry, Danish Center for Neonatal Screening, Department of Congenital Disorders, Statens Serum Institute, 2300 Copenhagen, Denmark; julc@ssi.dk; 5Section for Surgical Pathophysiology, Juliane Marie Center, Rigshospitalet, 2100 Copenhagen, Denmark; jesper.mehlsen.01@regionh.dk; 6Department of Clinical Physiology and Nuclear Medicine, Nordsjaellands Hospital, 2400 Hilleroed, Denmark; Louise.Schouborg.Brinth@regionh.dk

**Keywords:** myalgic encephalomyelitis, chronic fatigue syndrome, mitochondria, mitochondrial respiration, spare respiratory capacity, coupling efficiency, pyruvate dehydrogenase, bioenergetics, targeted metabolomics, large-scale discovery proteomics

## Abstract

Myalgic encephalomyelitis/chronic fatigue syndrome (ME/CFS) is a heterogeneous, debilitating, and complex disease. Along with disabling fatigue, ME/CFS presents an array of other core symptoms, including autonomic nervous system (ANS) dysfunction, sustained inflammation, altered energy metabolism, and mitochondrial dysfunction. Here, we evaluated patients’ symptomatology and the mitochondrial metabolic parameters in peripheral blood mononuclear cells (PBMCs) and plasma from a clinically well-characterised cohort of six ME/CFS patients compared to age- and gender-matched controls. We performed a comprehensive cellular assessment using bioenergetics (extracellular flux analysis) and protein profiles (quantitative mass spectrometry-based proteomics) together with self-reported symptom measures of fatigue, ANS dysfunction, and overall physical and mental well-being. This ME/CFS cohort presented with severe fatigue, which correlated with the severity of ANS dysfunction and overall physical well-being. PBMCs from ME/CFS patients showed significantly lower mitochondrial coupling efficiency. They exhibited proteome alterations, including altered mitochondrial metabolism, centred on pyruvate dehydrogenase and coenzyme A metabolism, leading to a decreased capacity to provide adequate intracellular ATP levels. Overall, these results indicate that PBMCs from ME/CFS patients have a decreased ability to fulfill their cellular energy demands.

## 1. Introduction

Myalgic encephalomyelitis/chronic fatigue syndrome (ME/CFS) is a heterogeneous, debilitating, complex, and controversial disorder with many uncertainties regarding both aetiology and delimitation towards other syndrome diagnoses such as fibromyalgia and postural orthostatic tachycardia syndrome (POTS) [1,2]. ME/CFS is characterised by extreme mental and physical fatigue and fatigability associated with symptoms of pain, instability in the control of a broad range of organ systems, tendency towards hypersensitive sense modalities, and inability to respond appropriately to stressors of all sorts causing post-exertional malaise [3]. As both pathophysiology and aetiology are undetermined and no definite biomarkers exist, the diagnosis of ME/CFS is based on exclusion criteria and symptoms [4,5]. ME/CFS is often found to be preceded by infections or prolonged extreme physiological or psychological strain [6]. The societal importance and impact of ME/CFS are considerable with an estimated prevalence of 0.2–0.8% [7,8], with symptom severity comparable to other chronic and severe conditions such as depression, multiple sclerosis, and stroke [9,10], and as few as 6% of diagnosed individuals returning to premorbid function [6].

The lack of strict and univocal diagnostic criteria coupled with marked heterogeneity and lack of valid biomarkers has hampered diagnosis, research, and comparison of studies. Both the aetiology and pathophysiology of ME/CFS remain undetermined. However, in the last decade, several studies have shown diverse biological markers being significantly affected in ME/CFS patients revealing a disease state characterised by sustained autonomic nervous system (ANS) dysfunction, inflammation, altered metabolism, and mitochondrial dysfunction [3]. The alteration observed in the ANS of ME/CFS patients is indicative of increased sympathetic and reduced parasympathetic regulation, especially autonomic regulation of the cardiovascular system [11,12,13,14,15,16,17]. The resulting orthostatic intolerance (OI) is a core symptom of ME/CFS and is mirrored in the frequent comorbidity of ME/CFS and POTS [2]. Moreover, studies have described the presence of G-protein coupled autoantibodies with agonistic effects on neuroendocrine receptors in the ANS in both ME/CFS and in associated conditions such as POTS and chronic regional pain syndrome (CRPS) [18,19,20,21]. The neuroendocrine receptors mediate the control of ANS through crosstalk with mitochondrial function and metabolism [22,23,24,25].

ME/CFS patients have shown to be metabolically reprogrammed with significant deviations in pathways affecting sphingolipids, phospholipids, purines, and amino acids with impaired metabolic responses to environmental stressors centred around mitochondrial function [26,27,28,29,30]. Moreover, studies have shown dysfunction of mitochondrial bioenergetics and regulations in mitochondrial respiratory chain complexes to compensate for inefficient ATP production [31]. Therefore, abnormal mitochondrial energy metabolism has become an area of interest in ME/CFS research in recent years [31,32,33,34]. In this study, we aim to perform deep profiling of the mitochondrial function and evaluate its association with symptom burden in ME/CFS through a case–control design with a well-defined cohort of patients that have developed post-viral ME/CFS. Mitochondrial function was evaluated in peripheral blood mononuclear cells (PBMCs) that have shown their capabilities to assess systemic changes in mitochondrial function in other chronic diseases and the correlation between mitochondrial function and physical and cognitive dysfunction [35,36,37,38].

## 2. Materials and Methods

### 2.1. Study Design

Six female patients between 30 and 50 years old were enrolled in the study, diagnosed with ME/CFS by a ME/CFS specialist, and recruited through the Danish ME association and with a mean illness duration of 12 years (Appendix A). Participants, patients and controls were recruited with the following characteristics: female, age 30–50 years old, and non-smokers. The exclusion criteria for the participants were: known acute or chronic illness, presence of autoimmune disease, and known mental illness. During recruitment, all participants were asked about exercise and lifestyle to avoid excessive differences in physical activity between patients and controls. The patients were asked to abstain from medications—including supplements—a week prior to the examination. All the participants, patients and controls, were tested during 2016 and 2017. Blood samples were collected 5 hours after a light breakfast. Participants had abstained from medication intake and extreme exercise for the previous week. Standard blood tests were conducted to discard possible differential diagnosis. The questionnaires were answered within 2 months of the blood samples collection. The patients were submitted to a narrative, clinical interview and schematic questioning using a checklist of symptoms filled in by the patients at home and by the staff at the clinic to elucidate symptom burden and onset. The onset of symptoms was mainly after viral infections. Patients 1 and 6 described acute onset of symptoms after tonsillitis and a combination of physical and psychological stress, respectively. Patients 4 and 5 described subacute onset following mononucleosis and episodes of pyelonephritis, respectively. Finally, patients 2 and 3 described insidious symptoms after several viral infections. These evaluations were supplemented with three self-assessed questionnaires of well-being: Fatigue Scale of Motor and Cognitive function (FSMC), Medical Outcomes Study 36-Item Short-Form Health Survey (SF-36), and COMPASS-31. FSMC evaluates fatigue symptoms by providing a global fatigue severity score as well as subscales of motor and mental fatigue [39]. SF-36 is a questionnaire that evaluates the health-related quality of life [40]. COMPASS-31 specifies and quantifies symptoms and severity of autonomic dysfunction [41].

### 2.2. Isolation of Human PBMCs Using a Vacutainer Cell Preparation Tube

Blood samples were collected by venipuncture in 8 mL citrate-containing Vacutainer cell preparation tubes (CPT) (Becton, Dickinson and Company, Frankline Lakes, NJ, USA) (Appendix A). PBMCs were isolated by differential centrifugation. The PBMC pellet was carefully resuspended in ammonium–chloride–potassium (ACK) lysis buffer (Gibco, Life Technologies Corporation, Grand Island, NY, USA) and incubated for 10 min to lyse any remaining red blood cells. Then, the PBMC pellet was washed with PBS (Sigma Aldrich, Steinheim, Germany) by centrifugation and was finally resuspended in Seahorse assay media (Agilent Technologies, Cedar Creek, TX, USA). The PBMCs yield and viability were determined using via-1 cassettes™ in image cytometer NC-3000 (Chemometec, Allerod, Denmark) according to the manufacturer’s instruction. The PBMCs viability threshold for inclusion was 90%. Cells were used immediately after isolation for bioenergetics analysis or frozen at −80 °C for further analysis.

### 2.3. Bioenergetics Profile

PBMCs (80,000 cells/well) were seeded in Seahorse 96-well (Seahorse XFe96, Agilent Technologies, MA, USA) pre-coated plates with 0.1 mg/mL Cell-tak™ (Corning, Discovery Labware Inc, Bedford, MA, USA). Media was freshly prepared with XF base media (Agilent Technologies, Cedar Creek, TX, USA) (pH 7.4) supplemented with 10 mmol/L glucose, 2 mmol/L sodium pyruvate, and 2 mmol/L glutamine (Sigma Aldrich, Hamburg, Germany). The bioenergetic profile comprised a total of 14 parameters (Appendix A). The Oxygen Consumption Rate (OCR) and Extracellular Acidification Rate (ECAR) were measured simultaneously in three cycles of mixing (3 min) and measuring (3 min) for each section: basal, injection 1, and injection 2. Three different groups of sequential injections were performed: (A)1 μmol/L Oligomycin (inhibitor of complex V); 2 μmol/LM Rotenone/2 μM Antimycin (inhibitor of complex I and complex III, respectively);(B)6 μmol/L FCCP (uncoupler); 2 μmol/L Rotenone/2 μM Antimycin;(C)1 μmol/L Oligomycin; 50 mmol/L 2-Deoxy-D-glucose (2-DG) (inhibitor of glycolysis).

The bioenergetic profile consists of 14 parameters obtained by combining the OCR and ECAR levels in basal conditions and in response to the different inhibitors; refer to Appendix A for a schematic representation of the bioenergetics profile analysis. The parameters and their formulas are described in Appendix A. The different bioenergetic parameters were calculated for each well. Then, the replicates were pooled for each bioenergetic parameter to calculate the mean and the standard deviation. 

To account for inter-experiment variations, we calibrated the Seahorse instrumentation before every experiment to account for day-to-day variation. Moreover, an inter-experiment blood sample from a healthy individual not related to this study was drawn on the same day as this study participants and was prepared and analysed simultaneously. We used this sample to assess overall assay performance, and it was not included in the bioenergetics profile calculations of the individuals in this study.

### 2.4. Protein Carbonyl Content

The amount of protein carbonylation was determined using the Oxiselect protein carbonyl ELISA kit from Cell Biolabs (San Diego, CA, USA), according to the manufacturer’s protocol. Briefly, each sample was analysed by triplicate with 1 μg of total protein lysate in each well. The proteins were adsorbed to the plate overnight at 4 °C, followed by washing and incubation with DNPH (2,4-Dinitrophenylhydrazine) at 0.04 mg/mL for 45 min. The excess DNPH was removed by washing with a mix of PBS/ethanol (1:1, *v*/*v*). The anti-DNP (2,4-dinitrophenyl) antibodies and the horseradish peroxidase-conjugated antibodies were incubated for 1 h with washing steps in between. The plate was read at 450 nm in a Synergy H1 microplate reader (Biotek, Winooski, VT, USA) using reduced BSA as a blank.

### 2.5. Protein Extraction and Sample Preparation

The frozen PBMC pellets were resuspended in 50 mmol/L HEPES lysis buffer (pH 7.4) with 0.3% SDS and protease-inhibitor cocktail tablets (Roche Diagnostic GmbH, Mannheim, Germany). The protein suspension was treated with ultrasonication (Branson Sonifier 250, Branson ultrasonics corp, Brookfield, CT, USA) on ice water, at output control 3 and 30% duty cycle for 3 rounds of 3 pulses with 1 minute on ice between each round. Insoluble proteins were removed by centrifugation at 16,000× *g* for 10 min at 4 °C.

### 2.6. Tandem Mass Tag Labelling, Isoelectric Focusing Separation, and Purification of Peptides

Equal protein amounts (80 μg) were processed according to the tandem mass tag (TMT) 11-plex manufacturer’s instructions (Thermo Fisher Scientific, Rockford, IL, USA). Briefly, each sample was sequentially reduced and alkylated with 200 mmol/L TCEP (tris(2-carboxyethyl)phosphine hydrochloride) and 375 mmol/L iodoacetamide, respectively. Then, the proteins were precipitated with acetone (−20 °C, overnight), followed by digestion with 2.5 μg trypsin. The peptides were labelled with different TMT labels for each sample. After TMT-labelling, the peptide samples were combined and subsequently purified using a strong cation exchange (Strata SCX, 55 µM, 70A, 100 mg/mL, Phenomenex, Torrence, CA, USA). The peptides were eluted with a mixture of 5% of ammonia and 30% methanol (Merck, Darmstadt, Germany) and subsequently vacuum-dried. The peptides were fractionated by isoelectric focusing (IEF) on a Multiphor II-unit (Pharmacia Biotech AB, Bromma, Sweden) using an 18 cm Immobiline Drystrip Gradient (IPG) pH 3–10 gel (GE Healthcare, Uppsala, Sweden). The sample was dissolved in rehydration solution containing 8 mol/L urea, 0.5% IPG buffer 3–10 (GE Healthcare, Uppsala, Sweden) and 0.002% bromophenol blue. The strip was then rehydrated with the sample at room temperature overnight. IEF was run at 59 kVh with the following program: 1 min gradient from 0–500 V, 1.5 h gradient from 500–3500 V followed by 16 h at 3500 V. The gel strip was cut in 10 pieces and peptides were extracted in three steps, of 1 hour each, with 100 μL 5% acetonitrile (AcN) (LC-MS grade, Merck, Darmstadt, Germany), 0.5% trifluoracetic acid (TFA) (Sigma Aldrich, Hamburg, Germany), and purified on PepClean C-18 Spin Columns (8 mg C18 resin, Pierce, Rockford, IL, USA) according to manufacturer’s protocol.

### 2.7. Nano-Liquid Chromatography and Mass Spectrometry (MS) Analysis

Each of the purified peptide mixtures was analysed twice by nano Liquid-Chromatography (nLC) (Easy-nLC 1000, Thermo Fisher Scientific, San Jose, CA, USA) coupled to a mass spectrometer (Q Exactive Plus, Thermo Fisher Scientific, Bremen, Germany) through an EASY-Spray nano-electrospray ion source (Thermo Fisher Scientific, Bremen, Germany). Pre-column (Acclaim PepMap 100, 75 µm × 2 cm, Nanoviper) and analytical column (EASY-Spray column, PepMap RSLC C18, 2 µm, 100 Å, 75 µM × 25 cm), both from Thermo Fisher Scientific (Vilnius, Lithuania) were used to trap and separate peptides using a 170-min gradient (5–40% Acetonitrile, 0.1% Formic acid). The MS was operated in positive mode and higher-energy collisional dissociation (HCD) with collision energy (NCE) of 32 for peptide fragmentation. Full-scan (MS1) resolution was 70,000, and AGC target set at 1 × 10^6^ with scan range between 391 and 1500 m/z. Data-dependent analysis (DDA) was applied to fragment up to 10 of the most intense peaks in MS1. Resolution for fragment scans (MS2) was set at 35,000 with first fixed mass at 121 m/z and AGC target at 2 × 10^5^. Dynamic exclusion was set at 19 seconds, and unassigned and +1 charge states were excluded. Furthermore, peptides with more than 10 peptide spectrum matches (PSMs) in the first analysis were excluded from the second analysis.

### 2.8. Proteomic Database Search and Statistical Analysis

The final database search was conducted in Proteome Discoverer 2.1 (Thermo Fisher Scientific, Bremen, Germany) with Mascot (Matrix Science, London, UK) on all raw files merged. Swiss-Prot was used as database with maximum two missed cleavages using trypsin as enzyme, and taxonomy was set for *Homo sapiens* (20,350 reviewed sequences from uniprot.org, September 2019). Precursor and fragment mass tolerance were set at 10 ppm and 20 mmu, respectively. Oxidation of methionine was set as dynamic modification, and static modifications were carbamidomethylation of cysteines and TMT-labels on lysine and peptide N-terminus. Co-isolation threshold set at 50%. Identification false discovery rate was set to 0.01. Proteins with a minimum of two unique peptides and five quantitative scans were included for further statistical analysis. The mitochondrial proteins were determined using the Mitocarta list with a 7% false discovery rate [42]. The proteomics data were filtered to obtain differentially altered proteins (DAPs) passing two criteria: fold change (FC) > 1.12 and <0.89 and student *t*-test *p*-value < 0.05. The FC criterion was established based on two times the median of the coefficient of variation of the control group. The group *p*-value of the pathways was performed by student *t*-test of the proteins involved in each pathway.

### 2.9. Targeted Metabolomics Profiling

Targeted metabolomics analysis of 408 metabolites in plasma samples was carried out using the AbsoluteIDQ^®^ p400 Kit (Biocrates Life Sciences AG, Innsbruck, Austria) and following the manufacturer’s instructions. Details on our analysis equipment and materials have been published previously [43]; additional details are available in Appendix A. Of the 408 metabolites, 42 were acquired by liquid-chromatography (LC) coupled with high-resolution mass spectrometry (HRMS), namely, the biogenic amines and the amino acids, and 366 were acquired by flow-injection analysis (FIA) coupled with HRMS, namely, the acylcarnitines, glycerophospholipids, sphingolipids, hexoses, cholesterol esters, and glycerides (details in Appendix A).

Plasma samples were aliquoted in cryotubes and stored at −80 °C until analysis. Samples were extracted according to instructions (10 µL) and injected in a randomised order on the same analytical plate and along with a 7-point calibration curve (LC-HRMS only), three blanks, and three quality control levels (QC1-3), of which QC1 and QC3 in singlicates and QC2 in triplicates. The normalisation of results using the median value of the QC2 replicates in comparison with their target values (as recommended by the manufacturer) was performed using MetIDQ Carbon-2793 software (Biocrates Life Sciences AG, Innsbruck, Austria).

We performed extensive quality control and filtering of the measurements using the MeTaQuaC R-based package v0.1.30 [43,44] (RStudio 1.3.1093, R 4.0.3) independently for LC and FIA measurements (as recommended). The automatically generated reports detail all parameters used and are available in the Appendix A. Preprocessed data (section 3.9 of the ME-CFS_MeTaQuaC_biocrates_qc_p400 reports in the Appendix A) were exported for further analysis. Statistical analyses were performed using MetaboAnalystR3.0.3 [45] in RStudio (see Appendix A). Missing values (1.2% after preprocessing) were replaced by 1/5 of the minimal positive values of their corresponding variables. Concentrations were further glog (generalised logarithm) transformed and Pareto scaled. Analyses included FC analysis, *t*-tests (false discovery rate, FDR, correction), principal component analysis (PCA), and hierarchical clustering heatmap. All original R scripts are available on https://github.com/SSI-Metabolomics/ME-CFS__SupplementaryMaterial (Accessed on: 4 January 2021).

### 2.10. Acylcarnitines and Organic Acids Profile

The plasma samples consist of plasma isolated by centrifugation from EDTA blood. For the measurement of amino acids, we first measured the summed molarity of L-amino acids (termed total amino acids) by an enzyme-linked immunosorbent assay (ELISA) (catalogue no. ab65347; Abcam, Cambridge, UK). This kit determines concentrations of free l-amino acids but not protein-bound or D-amino acids. Second, we identified and quantified each of the free amino acids by using the MassTrak Amino Acid Analysis (AAA) Solution Kit, an Acquity UPLC system equipped with a C18 BEH column (1.7 µm; 2.1 × 150 mm) and an integrated photodiode array detector operating at 260 nm (all from Waters Corporation, Milford, MA, USA). The samples were prepared according to the manufacturer’s instructions. Briefly, plasma was deproteinised with an equal amount of sulphosalicylic acid (10%) containing norvaline as internal standard and thoroughly mixed. After centrifugation, part of the supernatant was alkalised with a borate buffer/NaOH solution, derivatised with 6-aminoquinolyl-N-hydroxysuccinimidyl carbamate (AQC), and analysed by UHPLC-UV. UHPLC conditions were in accordance with the manufacturer’s instructions, except: (1) instead of MassTrak AAA Eluent B, we used AccQ-Taq Ultra Eluent B (Waters) to enhance separation of amino acid eluting midway in the chromatogram (1); and (2) a steeper gradient curve between 2 and 5.5 min to enhance separation of arginine and glycine [46]. Further details are available upon request.

The plasma concentration of free carnitine and 15 short-chain acylcarnitines, 6 medium-chain acylcarnitines, and 14 long-chain acylcarnitines were measured by ultra-HPLC tandem MS [47]. In short, 250 mL of plasma was mixed with deuterium-labelled internal standard and deproteinised with 700 mL of AcN. After centrifugation, the supernatant was evaporated and resuspended in 100 mL 1:3 methanol/solvent A. We injected a 2 mL sample on an Acquity UPLCTM (Waters, Milford, MA, USA) equipped with an Acquity UPLCTM BEH C18 column (2.1 3 100 mm, 1.7 mm), using a flow rate of 0.35 mL/min and kept at 40 °C. The elution mobile phase consisted of a linear gradient using 0.2% heptafluorobutyric acid in 10 mM ammonium formate (solvent A) and 100% methanol (solvent B): 2.5–20% solvent B for 0.34 minutes, 20–90% solvent B for 18 min, and 90–100% solvent B for 1 min. Multiple reaction monitoring was performed on a Quattro MicroTM Tandem Mass Spectrometer (Waters Corporation, Milford, MA, USA) with an electrospray ionisation interface under positive-ion detection mode. The interface heater was maintained at 300 °C, and ion spray voltage was 3 kV. The dissolving gas (nitrogen), cone gas (nitrogen), and collision-activated dissociation gas (argon) were set at 700 L/h, 50 L/h, and 10 psi, respectively. The dwell time for each analyte was 75 ms. Optimisation of mass transitions and collision energy was performed by flow injection analysis of standard solutions.

### 2.11. Steroid Analysis

We used the targeted LCMS CHS MSMS Steroids Kit (PerkinElmer Inc., Waltham, MA, USA) to measure the concentration of 17-hydroxyprogesterone, testosterone, androstenedione, and cortisol in the plasma samples. More detailed methods are in Appendix A.

### 2.12. Statistical Analysis

All results are presented as means ± standard deviation (SD) unless stated otherwise. The patients’ clinical characteristics and steroid plasma levels were compared by student *t*-test for the continuous variables. Bioenergetics values were compared between the two groups by *t*-test with Bonferroni correction for multiple comparisons. Bivariate correlation analysis was performed using Pearson’s r coefficient. Statistical analysis was performed using R studio version 1.2.5033 [48] and Excel (Microsoft, Redmond, WA, USA).

## 3. Results

### 3.1. Description of the Cohort

All patients enrolled in the study were evaluated for the following diagnostic criteria for myalgic encephalomyelitis/chronic fatigue syndrome (ME/CFS): Fukuda Criteria [5], Revised Canadian Criteria [49], International Consensus Criteria (ICC) [50], and American Institute of Medicine (IOM) criteria [3] (Appendix A). All six patients fulfilled the Fukuda Criteria, five the ICC and OIM criteria, and four the Revised Canadian Criteria. Patients described an average disease duration of 12 years (range 7–21 years) and did not differ in relation to standard demographic data such as age, height, weight, BMI, blood pressure, and heart rate (Table 1). Table 2 shows a summary of the results from the questionnaires. In the FSMC questionnaire, patients showed significantly increased mental and physical fatigue levels compared to controls. In the SF-36 questionnaire, the patients scored lower than the controls on physical-, but not emotional-related parameters. In the COMPASS-31 questionnaire, patients scored significantly higher than healthy controls, mirroring a severe autonomic nervous system (ANS) dysfunction. Bivariate analysis of these parameters showed a positive correlation between FSMC total score and COMPASS-31 weighted score (r = 0.93, *p*-value 3.47 × 10^−4^) (Figure 1A). In contrast, FSMC total score correlated negatively with SF-36 physical functioning (r = −0.96, *p*-value = 4.10 × 10^−5^) (Figure 1B), as well as other SF-36 scores, except well-being, pain, and health change (Appendix A). COMPASS-31 also correlated negatively with different SF-36 scores, except well-being, pain, and health change (Appendix A). We also performed correlation analysis in the patients’ group to differentiate which of these results could be caused by the separation of the patients’ and controls’ group in the FSCM total score (Appendix A). The patients’ group showed significant positive correlation between FSCM total score and COMPASS-31. Overall, these results indicate that the physical health of the patients is connected to fatigue and ANS dysfunction.

Studies in ME/CFS have shown autoantibodies with agonistic effects on neuroendocrine receptors in the ANS. These receptors are directly connected to the ANS through dialogues between mitochondrial function and metabolism. Therefore, we focused on investigating abnormal mitochondrial energy metabolism in PBMCs from patients with ME/CFS.

### 3.2. Bioenergetics Profile 

We performed bioenergetics profiling to investigate alterations in energy metabolism as a possible driver for the patients’ symptoms. We analysed the functional capacity of the two major energy-producing pathways in PBMCs by recording the oxygen consumption rate (OCR) and extracellular acidification rate (ECAR), reflecting mitochondrial respiration and glycolysis towards lactate, respectively. Comparison of bioenergetics parameters showed no significant differences in glycolysis between patients and controls (Appendix A). However, we observed differences in mitochondrial respiration, specifically, coupling efficiency (FC = 0.87, *p*-value 0.041) (Figure 2A–F). No other differences were observed in the remaining parameters of bioenergetics profiling (Figure 2). Coupling efficiency correlated negatively with disease duration (r = −0.84, *p*-value = 0.005) and positively with SF-36 general health score (r = 0.86, *p*-value = 0.003) (Figure 2G,H). No significant correlation was observed between coupling efficiency and FSMC or COMPASS-31 (Appendix A).

### 3.3. Protein Carbonyl Content

The uncoupling of the mitochondrial respiratory chain is a well-known regulator of reactive oxygen species that may lead to oxidative stress and oxidative damage to biomolecules such as proteins. Protein carbonyl content was used to assess oxidative damage and showed a tendency to be higher in ME/CFS patients than controls (FC = 1.36, *p*-value = 0.36), indicating that PBMCs from ME/CFS patients could have higher levels of oxidative damage (Figure 2I). However, the variation in the PBMC protein carbonyl content, both in patients and controls (coefficient of variation of 0.52 and 0.46, respectively), indicates a large interindividual variation.

### 3.4. Proteomics Profile

To gain insights into the possible mechanisms of disturbed mitochondrial bioenergetics in ME/CFS, we performed large-scale discovery proteomics of PBMCs, yielding robust identification and quantification of more than 3300 proteins (Appendix A). To be considered differentially regulated, proteins had to pass both Student’s *t*-test and fold change criteria as described in Materials and Methods. Twenty-two proteins were found differentially altered proteins (DAPs) (13 increased and 9 decreased proteins, respectively) (Figure 3A) (Table 3).

The two most significantly altered proteins in PBMCs from ME/CFS patients are related to the immune system, cathepsin W (CTSW) (FC = 1.53), involved in the major histocompatibility complex (MHC) class I pathway, and human leukocyte antigen (HLA) protein histocompatibility antigen, alpha chain E, (HLA-E) (FC = 1.34) belonging to the MHC class Ib. In contrast, mixed lineage kinase domain-like protein (MLKL) (FC = 0.89), which plays a critical role in tumour necrosis factor (TNF)-induced necroptosis, was significantly lower in ME/CFS patients than controls, verall, indicating an alteration of the immune response in ME/CFS patients. The third most significantly altered protein was LAMTOR1 (FC = 1.32). LAMTOR1, together with mildly higher levels of LAMTOR5 (FC = 1.1, *p*-value = 0.03), reflect increased levels of the ragulator complex that recruits mammalian target of rapamycin complex 1 (mTORC1) to the lysosomes for activation. Overall, this indicates that mTORC1 could be more active in PBMCs from ME/CFS patients. We also observed alterations in protein levels that could indicate telomere shortening in PBMCs from ME/CFS patients. These proteins are C9orf78 (telomere length and silencing protein 1 homolog) (FC = 1.19), a telomeric heterochromatin assembly factor involved in telomere length maintenance [51]; and LAGE3 (EKC/KEOPS complex subunit LAGE3) (FC = 0.84), one of the five proteins conforming the kinase, endopeptidase, and other proteins of small-size (KEOPS) complex [52] that is linked to telomere length regulation [53,54]. 

Interestingly, PBMCs from ME/CFS patients showed lower levels in four mitochondrial proteins: PDPR (FC = 0.85), PANK2 (FC = 0.85), ATP5FIE (FC = 0.86), and SLC25A24 (FC = 0.88). However, there was no change in total mitochondrial protein levels (median FC = 0.99 of 279 mitochondrial proteins) (Figure 3B). Lower levels of PDPR (pyruvate dehydrogenase phosphatase regulatory subunit) could reflect a lower activity of the pyruvate dehydrogenase complex (PDH), linking non-mitochondrial glycolysis with mitochondrial ATP production. Moreover, lower levels of PANK2 (panthothenate kinase 2) (FC = 0.85) and VNN1 (pantetheinase, vascular non-inflammatory molecule-1) (FC = 0.75, *p*-value = 0.03) could mirror lower cellular CoA levels, leading to impaired mitochondrial energy production. In the same direction, lower amounts of AT5FIE (ATP synthase F1 subunit epsilon) could represent insufficient ATP synthase activity, leading to impaired ATP production. Moreover, lower levels of SLC25A24 (calcium-binding mitochondrial carrier protein SCaMC-1) can reflect an impaired exchange of ATP-magnesium and phosphate between mitochondria and the cytosol, leading to disturbed mitochondrial ATP production. Lastly, we observed higher levels of SLC16A3 (monocarboxylate transporter 4) (FC = 1.13, *p*-value = 0.028), which could relate to increased excretion of cellular lactate, secondary to the lack of mitochondrial ATP production. 

Even though proteomic and bioenergetic data supported a metabolic regulation of energy metabolism, the proteins of the central energy-producing metabolic pathways did not show altered levels (Appendix A). However, of the 209 proteins quantified from these pathways, four proteins showed minor but significant changes. Two of the proteins were lower in PBMCs from ME/CFS patients than controls; MCCC2 (subunit of 3-methylcrotonoyl-CoA carboxylase) involved in breaking down the amino acid leucine to acetyl-CoA, and SUCLG1, a subunit of the citric acid cycle enzyme, succinate coenzyme A ligase, which catalyses the conversion of succinyl-CoA to succinate. Two proteins were higher in PBMCs from ME/CFS patients than controls; HK2 (hexokinase2), which catalyses the rate-limiting and first obligatory step of glycolysis, and PCK2 (phosphoenolpyruvate carboxykinase 2), a mitochondrial enzyme that catalyses the conversion of oxaloacetate to phosphoenolpyruvate as a rate-limiting step in gluconeogenesis. Notably, PCK2 preferentially converts oxaloacetate derived from lactate and, thus, can support biosynthesis in glycolytic cells. Together with findings from the proteomic profile, these data indicate a rewiring of metabolism in patients’ PBMCs from mitochondrial oxidative phosphorylation to glycolysis with increased recycling of lactate to support cellular biosynthesis.

Finally, we analysed the proteomic profile of cluster of differentiation (CD) proteins (10 proteins), surface markers used for identifying PBMCs subtypes and evaluating whether changes in the distribution of the PBMC population caused the changes observed. The levels of CD proteins were similar in ME/CFS patients and controls (data not shown). To further evaluate the implications of the proteomic profile, we analysed the possible correlations with the other parameters measured. We found no correlation between the proteomic profile and the rest of the parameters (data not shown).

To conclude, based on the proteins’ functions, the top mechanisms altered in PBMCs from ME/CFS patients are immunity, telomere biology, mitochondrial metabolism, and mTORC1. Overall, these results were not explained by changes in mitochondria amount or PBMCs subtypes’ distribution in PBMCs from ME/CFS patients. The proteomic findings are summarised in Figure 4.

### 3.5. Targeted Metabolomics Profiling

Routine metabolic screening for inborn errors of metabolism (plasma acylcarnitines and urine amino and organic acids) was negative (data not shown). To further evaluate possible metabolic changes, we performed a more comprehensive targeted metabolic profiling in plasma from patients and controls. Quality control of the measurements showed that the kit performance was satisfactory (see the MeTaQuaC reports in Appendix A). Technical and biological compound variability (section 4.3.2 of the reports, see Appendix A) could be separated; however, the relation between the two was not entirely conclusive, probably due to the small sample size. After preprocessing, 181 compounds were included in statistical analyses, of which 27 and 154 acquired with LC-HRMS and FIA-HRMS, respectively. Boxplots of concentrations of each included compound in patients and controls are available in section 6.1.1.2.1 of the reports (see Appendix A).

A number of 25 compounds were outside the defined controls/patients fold-change thresholds of >1.3 or <0.77 (see Appendix A). However, univariate and multivariate statistical analyses revealed no compound significantly different between patients and controls (after false discovery rate correction for multiple comparisons, see Appendix A). Nonetheless, two compounds from the phosphatidylcholines class (PC(36:4) and PC-O(34:4)) seemed less abundant in ME/CFS patients than in controls, with FCs of 0.75 and 0.60, respectively, and raw *p*-values of 0.027 and 0.037, respectively (Figure 5).

### 3.6. Steroid Analysis

Steroid plasma levels were not significantly different between ME/CFS patients and controls (Appendix A).

## 4. Discussion

In this study, we describe a comprehensive molecular evaluation of a small cohort of ME/CFS patients presenting with severe fatigue (FSMC score > 63), both mental and physical. The FSMC scores are significantly higher than FSMC scores reported in patients with inherited mitochondrial diseases [55], patients with stroke [56], and multiple sclerosis [39]. Moreover, the information collected about functional status and well-being (SF-36) showed limitation due to physical health, not emotional problems (Table 3). Other studies have shown that the SF-36 scores for ME/CFS patients are lower than those of other chronic diseases such as type 2 diabetes, cardiovascular diseases, and multiple sclerosis [9,57]. Regarding symptom severity mirroring ANS dysfunction, the patients had a score 27 times higher than the controls in the COMPASS-31 questionnaire, which is comparable to patients diagnosed with ANS failure [58]. Interestingly, we found a correlation between FSMC score and COMPASS-31 score, suggesting that ANS dysfunction is a central driver of the patient’s fatiguing illness. Several of the symptoms evaluated by the COMPASS-31 can be explained by the possible dysregulation of autoantibodies directed against adrenergic and muscarinic receptors [18,20,21]. Adrenergic signalling results in changes in mitochondrial function and the production of reactive oxygen species (ROS) [59,60]. Future studies of a larger cohort should investigate the relationships between symptom score, autoantibodies, ANS, and mitochondrial dysfunction in ME/CFS.

The PBMCs bioenergetic profile of the ME/CFS patients showed a significantly lower coupling efficiency compared to controls in basal conditions. Coupling efficiency is a tightly regulated process that represents how much ATP is produced per molecule of oxygen. The lower coupling efficiency observed leads to a lower ATP production and could be related to the slightly higher proton leakage seen in the patients’ cells. We also observed a tendency of higher spare respiratory capacity reflecting how much of the maximal respiration is being used by the cells. This higher spare respiratory capacity could be a compensatory mechanism producing more mitochondrial ATP, especially under stress conditions such as higher physical or mental activity. However, we observed a tendency of decreased ATP-linked to maximal respiration, indicating that PBMCs from ME/CFS patients cannot produce as much ATP as the controls even with increased spare respiratory capacity. Similar to our study, immortalised lymphocytes from ME/CFS patients have shown increases in proton leakage and spare respiratory capacity with non-significant changes in basal respiration [31], hereas another study on PBMCs from ME/CFS patients has shown no differences in coupling efficiency and a lower spare respiratory capacity [34]. This discrepancy can be due to technical differences in the protocols used to measure mitochondrial respiration. This previous study used frozen PBMCs, while we used fresh PBMCs, and the third study used immortalised cells. These different procedures could lead to changes in mitochondrial respiration. Moreover, we used a modified protocol to avoid the sequential injections of oligomycin and FCCP that can lead to underestimation of the maximal respiration and spare respiratory capacity [61]. This incongruity can also be explained by the heterogeneity of the ME/CFS population and reflects the importance of replicating biomarker studies in independent patient cohorts to validate their diagnostic sensitivity or usefulness for subgrouping of patients in future research studies. 

To further evaluate these changes in mitochondrial respiration, we performed large-scale discovery proteomics. One of the causes of the changes in mitochondrial respiration can be changes in mitochondrial mass. However, the median of the 279 mitochondrial proteins quantified showed no differences. Nonetheless, when analysing the levels of mitochondrial proteins, we observed a significantly decreased level of PDPR that can lead to lower activation of the PDH. Pyruvate is metabolised from glucose in the cytoplasm and transported into the mitochondria by a specific transporter. Once in the mitochondrial matrix, it is metabolised by the PDH to acetyl-CoA to be incorporated into the citric acid cycle and mitochondrial respiration. In addition, we detected lower levels of VNN1 and PANK2, which are both involved in CoA biosynthesis and regeneration. VNN1 is responsible for the recycling of vitamin B5 [62], an essential nutrient and precursor of CoA. PANK2 phosphorylates vitamin B5 in the first step of CoA biosynthesis. CoA is used in the mitochondrial oxidation of glucose, fatty acids, and some amino acids to acetyl-CoA, the major input to the citric acid cycle and is, thus, essential for mitochondrial energy production. A potential decrease in PDH, VNN1, and PANK2 activities could lead to lower mitochondrial ATP production and increased lactate production from glycolysis. Accordingly, patients’ cells showed higher levels of SLC16A3, the monocarboxylate transporter 4 (MCT4), which is upregulated in lactate-exporting glycolytic cells [63], along with upregulation of phosphoenolpyruvate carboxykinase 2 (PCK2), which is involved in the recycling of lactate to support cellular biosynthesis in glycolytic cells [64,65]. Dysregulated pantothenic acid and CoA metabolic pathways have already been shown in ME/CFS patients [28,66]. Interestingly, reduced activation of the PDH has been reported in other studies, but by upregulation of the PDH kinase instead of downregulation of the PDH phosphatase [29]. Recent studies in muscle cells from ME/CFS patients point to a PDH dysfunction as a potential cause of mitochondrial dysfunction [67].

The higher levels of LAMTOR1 and LAMTOR5 indicate an upregulation of mTORC1 in PBMCs from ME/CFS patients [68]. Activation of mTORC1 initiates a signalling cascade leading to mitochondrial biogenesis and upregulation of the expression of mitochondrial respiratory chain proteins [69]. However, we observed no general upregulation of mitochondrial amount or respiratory chain proteins (Figure 3B,C). We observed, however, lower levels of ATP5F1E, a complex V subunit. Previous studies in immortalised lymphocytes from ME/CFS patients have shown complex V deficiency and hyperactivation of mTORC1 [31,70]. In addition, downregulation of SLC25A24 can also indicate disturbed ATP synthesis [71]. One can hypothesise that the upregulation of mTORC1 is a compensatory mechanism for the inefficient mitochondrial ATP production, hampered by a disturbed mitochondrial respiratory chain structure or electron flow, as reflected by the lower coupling efficiency. Future studies should investigate the integrity of the respiratory chain in ME/CFS.

PBMC changes in mitochondrial proteins were not reflected in plasma metabolites. However, we found two compounds from the phosphatidylcholines (PC) class (PC(36:4) and PC-O(34:4)) with nominally significant lower levels in the patient group (before correction for multiple comparisons). PCs are known to support many of the body’s functions, ranging from fat metabolism, maintaining cell structure, and regulating the neurotransmitter acetylcholine in the brain [72,73]. PCs have previously been found significantly downregulated in a larger cohort of ME/CFS patients [57].

The heterogeneity of ME/CFS, the lack of definite biomarkers, and the difficult delineation towards other syndromes hamper the research into molecular mechanisms of the disease. Thus, in this study, we aimed to recruit a homogenous group of ME/CFS patients and applied strategies for personalised medicine by performing many tests in a few individuals to shed light on molecular mechanisms of disease in these individuals. Despite the small sample size, our data indicate a dysregulated mitochondrial metabolism centred on PDH and CoA metabolism, which supports findings from other and larger ME/CFS cohorts [29]. Interestingly, between 35 and 40% of ME/CFS patients experience significant improvement in their health when treated with dichloroacetate, a well-known activator of PDH [74,75]. These open-label studies need further follow-up; however, together with the present study and the previous metabolomics and proteomic studies of independent ME/CFS cohorts, they support dysregulated mitochondrial metabolism as an important feature of ME/CFS pathology. To some degree, abnormalities in bioenergetics parameters of blood cells have shown to correlate with the severity of ME/CFS symptoms [31,76]. Similarly, coupling efficiency correlated with symptom severity and disease duration in our cohort of ME/CFS patients.

## 5. Conclusions

In conclusion, our study presents a comprehensive evaluation both at the physiological and cellular level of a small, well-described ME/CFS patient cohort. We observed a mitochondrial dysfunction centred on PDH and CoA metabolism, which supports findings from other and larger ME/CFS cohorts. Since a similar metabolic dysregulation has been linked to other pathological conditions [77,78,79,80,81]; additional studies are needed to identify the mechanisms and medical impact of PDH dysregulation in ME/CFS. The possible relationship between ANS autoimmunity and mitochondrial dysfunction in ME/CFS deserves further investigations.

## Figures and Tables

**Figure 1 biomolecules-11-00961-f001:**
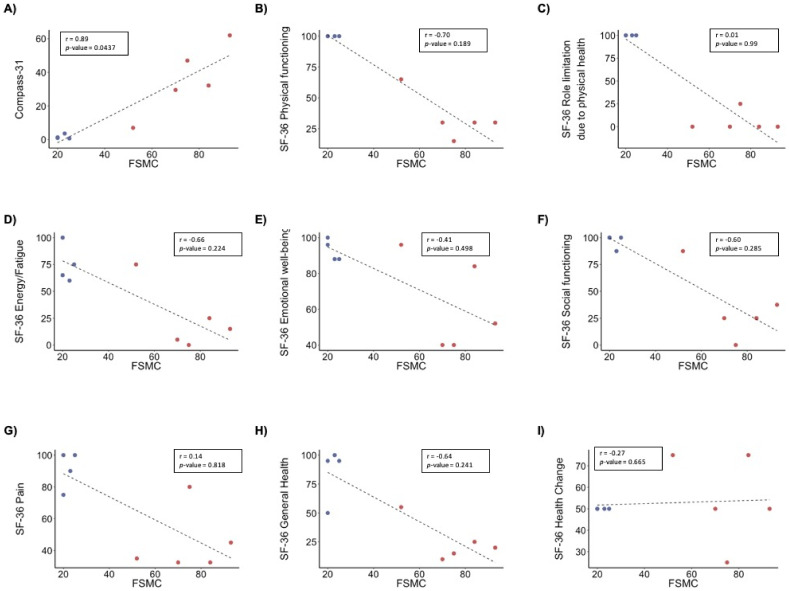
Correlation analyses of self-assessed fatigue in ME/CFS patients and controls. Pearson correlation analysis of FSCM total score with: (**A**) compass-31, (**B**) SF-36 physical functioning, (**C**) SF-36 role limitation due to physical health, (**D**) SF-36 energy and fatigue, (**E**) SF-36 emotional well-being, (**F**) SF-36 social functioning, (**G**) SF-36 pain, (**H**) SF-36 general health, and (**I**) SF-36 health change. p-values and Pearson (r) values were generated using Pearson’s correlation analysis. n = 10 (4 controls, 6 ME/CFS patients). Blue dots, controls; red dots, ME/CFS patients. n = 9 (4 controls, 5 ME/CFS patients).

**Figure 2 biomolecules-11-00961-f002:**
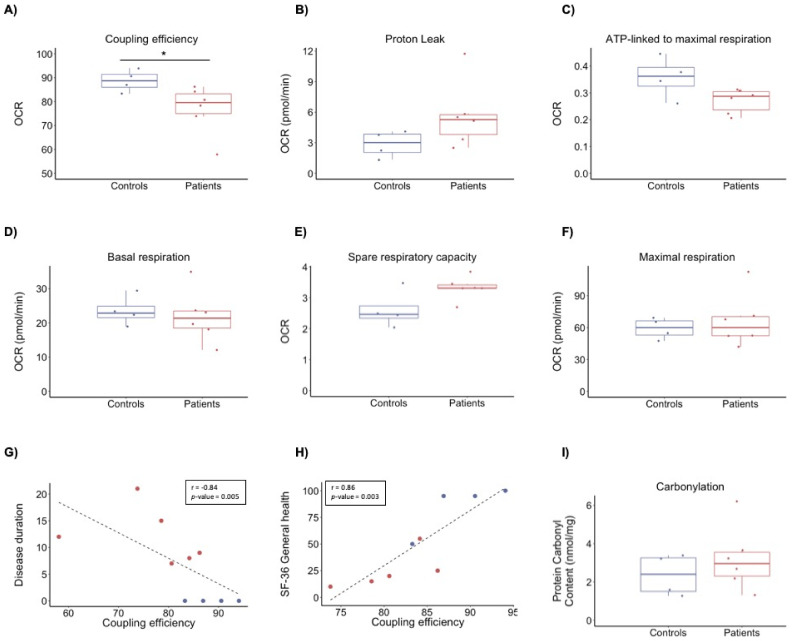
Bioenergetics profile of PBMCs from ME/CFS patients and healthy individuals as controls. Rates of mitochondrial respiration were measured in PBMCs using the Seahorse XFe96 instrument with added glucose, pyruvate, and glutamine. (**A**) Coupling efficiency (%) calculated as the ratio of ATP-linked respiration to basal respiration. (**B**) Proton leak calculated after the addition of oligomycin. (**C**) ATP-linked to maximal respiration calculated as the ratio of ATP-linked respiration to maximal respiration. (**D**) Basal mitochondrial respiration rates calculated before the addition of inhibitors. (**E**) Spare respiratory capacity calculated as the ratio of maximal respiration to basal. (**F**) Maximal respiration calculated after uncoupling of the respiratory chain by the addition of FCCP. (**G**) Pearson’s correlation between disease duration (y-axis) and coupling efficiency (x-axis) for all 10 individuals. (**H**) Pearson’s correlation between SF-36 general health (y-axis) and coupling efficiency (x-axis) for 9 individuals. The data of one patient lacks due to incompletion of the questionnaires. (**I**) Box plots representing the cellular levels of oxidative protein damage. Blue bar, controls; red bar, ME/CFS patients. n = 10 (4 controls, 6 ME/CFS patients). *p*-values denote significance from Student’s t-test. * *p* < 0.05.

**Figure 3 biomolecules-11-00961-f003:**
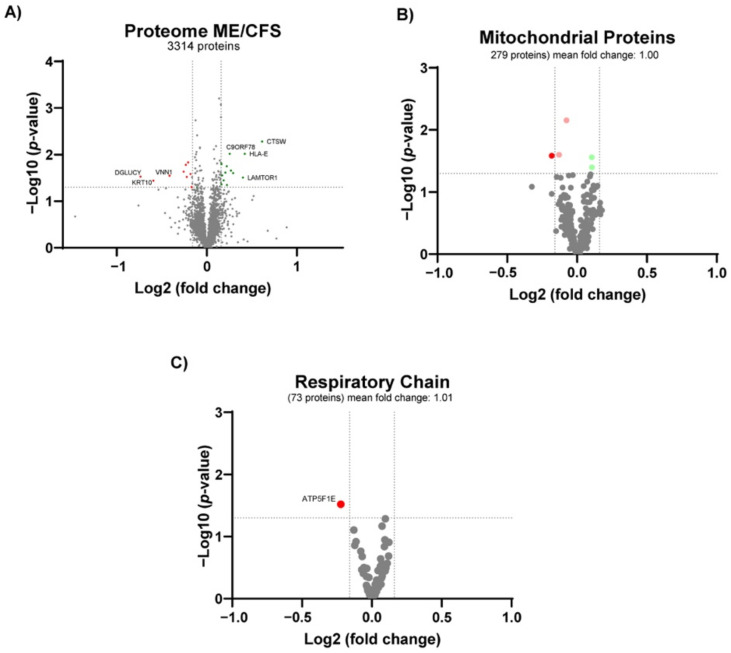
Proteomics profile of PBMCs from ME/CFS patients and healthy individuals as controls. Volcano plot of (**A**) all the quantified proteins, (**B**) the mitochondrial proteins and (**C**) respiratory chain proteins. n = 10 (4 controls, 6 ME/CFS patients). Red dots denote significantly lower proteins. Green dots denote significantly higher proteins. Light red and green dots denote significantly regulated proteins with a fold-change below the cut-off.

**Figure 4 biomolecules-11-00961-f004:**
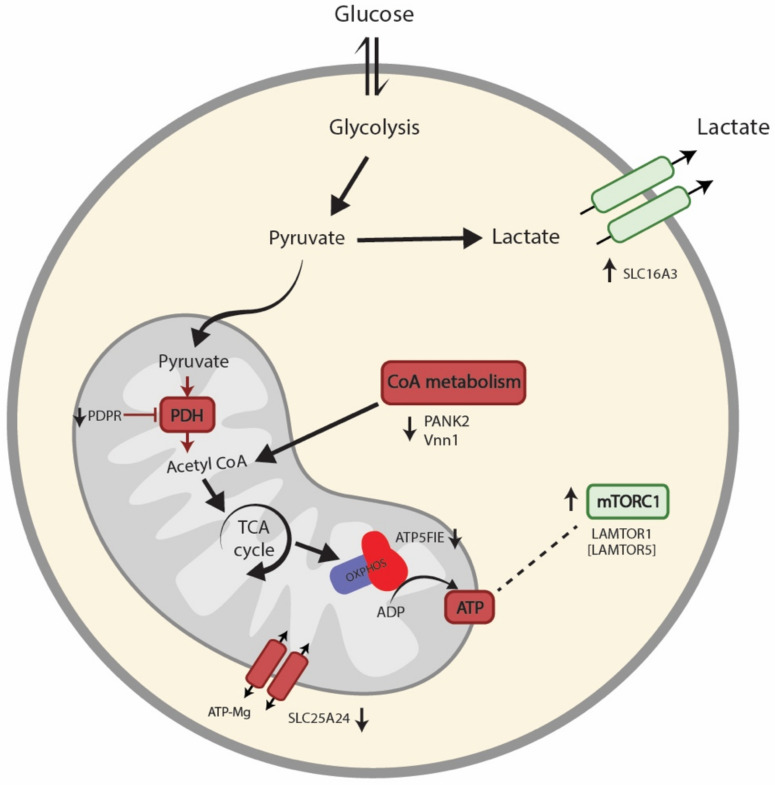
Summary of proteomics findings in PBMCs from ME/CFS patients and healthy individuals. We found reduced expression levels of PDPR, which can lead to lower activation of the PDH. In addition, downregulation of VNN1 and PANK2 could indicate lower cellular CoA levels. Thus, a potential decrease in PDH, VNN1, and PANK2 activities could lead to decreased mitochondrial ATP production and increased lactate production from glycolysis. Accordingly, the upregulation of SLC16A3 has been shown in lactate-exporting glycolytic cells. Further, we found downregulation of ATP5F1E and SLC25A24, which can suggest impaired ATP synthesis. The increased levels of LAMTOR1 and LAMTOR5 observed indicate an upregulation of mTORC1, which could be a compensatory mechanism for the lack of ATP in PBMCs from ME/CFS patients.PBMCs: peripheral blood mononuclear cells. ME/CFS: myalgic encephalomyelitis/chronic fatigue syndrome. PDPR: pyruvate dehydrogenase phosphatase regulatory subunit. PDH: pyruvate dehydrogenase complex. VNN1: Pantetheinase, vascular non-inflammatory molecule-1. PANK2: Panthothenate kinase 2. CoA: coenzyme A. ATP: adenosine triphosphate. SLC16A3: monocarboxylate transporter 4. AT5FIE: ATP synthase F1 subunit epsilon. SLC25A24: calcium-binding mitochondrial carrier protein SCaMC-1. mTORC1: Mammalian target of rapamycin complex 1. LAMTOR 1 and 5: ragulator complex protein LAMTOR1/LAMTOR5.

**Figure 5 biomolecules-11-00961-f005:**
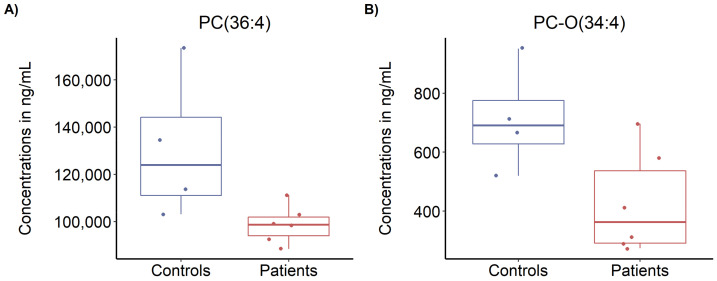
Targeted metabolomics profiling. (**A**) Levels of PC(36:4), and (**B**) levels of PC-O(34:4) in plasma from ME/CFS patients and controls. Blue bar, control; red bar, ME/CFS patients. n = 10 (4 controls, 6 ME/CFS patients).

**Table 1 biomolecules-11-00961-t001:** Demographic characteristics of the participants. Data are represented as mean ± SD.

Characteristics	Controls (n = 4)	ME/CFS Patients (n = 6)
Age (years)	42 ± 11	39 ± 5
Height (cm)	173 ± 6.8	169 ± 4.7
Weight (kg)	65.5 ± 4	60.3 ± 4.5
BMI (kg/m^2^)	21.9 ± 1.6	21.2 ± 1.2
Systolic blood pressure (mmHg)	103 ± 12	110 ± 9
Diastolic blood pressure (mmHg)	69 ± 5	73 ± 9
Heart Rate (bpm)	66 ± 2	69 ± 6
Disease duration (years)	NA	12 ± 5.3

**Table 2 biomolecules-11-00961-t002:** Clinical questionnaires.

Characteristics	Controlsprekidanje(n = 4)	ME/CFS Patientsprekidanje(n = 6)
FSCM	22 ± 2.4	74.8 ± 15.5 ***
FSCM Mental fatigue	11.8 ± 2.1	37 ± 8.9 **
FSCM Physical fatigue	10.3 ± 0.5	37.8 ± 7.2 ***
SF-36 Physical functioning	100 ± 0	34 ± 18.5 ***
SF-36 Role limitations due to physical health	100 ± 0	5 ± 11.2 ***
SF-36 Role limitations due to emotional problems	100 ± 0	100 ± 0
SF-36 Energy/fatigue	75 ± 18	24 ± 30.1 *
SF-36 Emotional well-being	93 ± 6	62 ± 26
SF-36 Social functioning	97 ± 6	35 ± 32.4 *
SF-36 Pain	91 ± 12	45 ± 20.2 **
SF-36 General health	85 ± 23	25 ± 17.7 **
SF-36 Health change	50 ± 0	55 ± 20.9
Compass-31	1.6 ± 1.3	35.5 ± 20.6 *

Fatigue Scale of Motor and Cognitive function (FSMC) evaluates symptoms of fatigue by giving a total fatigue score severity as well as sub-scores of mental and motor fatigue [39]. The Medical Outcomes Study 36-Item Short-Form Health Survey (SF-36) evaluates health-related quality of life [40], and COMPASS-31 specified and quantifies symptoms and severity of autonomic dysfunction [41]. Data are represented as mean ± SD * *p*-value < 0.05, ** *p*-value < 0.01, *** *p*-value < 0.001.

**Table 3 biomolecules-11-00961-t003:** Proteomic analysis with differentially altered proteins (DAPs) in the ME/CFS patients compared to the controls.

Accession Number	Description	Gene Name	Fold Change	*p*-Value
P56202	Cathepsin W	CTSW	1.53	0.005
P13747	HLA class I histocompatibility antigen, alpha chain E	HLA-E	1.34	0.010
Q6IAA8	Ragulator complex protein LAMTOR1	LAMTOR1	1.32	0.031
Q9BYM8	RanBP-type and C3HC4-type zinc finger-containing protein 1	RBCK1	1.22	0.025
Q9BSJ2	Gamma-tubulin complex component 2	TUBGCP2	1.20	0.022
Q9NZ63	Telomere length and silencing protein 1 homolog	C9orf78	1.19	0.010
O75718	Cartilage-associated protein	CRTAP	1.17	0.045
Q9NYM9	BET1-like protein	BET1L	1.17	0.018
Q96DM3	Regulator of MON1-CCZ1 complex	RMC1	1.15	0.024
O95372	Acyl-protein thioesterase 2	LYPLA2	1.14	0.036
O15427	Monocarboxylate transporter 4	SLC16A3	1.13	0.028
Q8TCD5	5’(3’)-deoxyribonucleotidase, cytosolic type	NT5C	1.12	0.043
P49459	Ubiquitin-conjugating enzyme E2 A	UBE2A	1.12	0.016
Q8NB16	Mixed lineage kinase domain-like protein	MLKL	0.89	0.049
Q6NUK1	Calcium-binding mitochondrial carrier protein SCaMC-1	SLC25A24	0.88	0.026
Q8NCN5	Pyruvate dehydrogenase phosphatase regulatory subunit, mitochondrial	PDPR	0.87	0.015
P56381	ATP synthase subunit epsilon, mitochondrial	ATP5F1E	0.86	0.030
Q9BZ23	Pantothenate kinase 2, mitochondrial	PANK2	0.85	0.017
Q14657	EKC/KEOPS complex subunit LAGE3	LAGE3	0.84	0.023
O95497	Pantetheinase	VNN1	0.75	0.028
P13645	Keratin, type I cytoskeletal 10	KRT10	0.66	0.036
Q7Z3D6	D-glutamate cyclase, mitochondrial	DGLUCY	0.60	0.030

## Data Availability

All data generated and analysed during this study are included in this published article (and its Appendix A), except for individualised data on diagnostic parameters due to data protection regulations.

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
