# Peer review of "Bioenergetic and Proteomic Profiling of Immune Cells in Myalgic Encephalomyelitis/Chronic Fatigue Syndrome Patients: An Exploratory Study"

_biomolecules, 2021, doi:10.3390/biom11070961_

Round 1
Reviewer 1 Report
The study is clearly described, and the studies are carefully documented. The major problem is that only 6 patients and 4 controls are tested, rally too small to even be considered a cohort. Nonetheless, carefully conducted studies on even small numbers of ME/CFS patients are of interest to the field. In a reply to a question about measurement error and stability of findings, the authors describe this as an exploratory study. That is an excellent description of the situation, and the term should be included in the title, i.e. “Bioenergetic and proteomic profiling of immune cells in ME/CFS patients – An exploratory study”. The data are encouraging and will be useful to prompt larger studies to explore this question more fully. However, the findings cannot be considered definitive. Abstract uses term “well-defined cohort” (line 25). The patients are well characterized clinically, but well-defined cohort is misleading. The numbers of patients and controls need to be included in the abstract to temper findings. The authors state that the study recruited a homogeneous group of ME patients. However little specific information about the patient’s illness, other than indicating case definition fulfilled. The heterogeneity of ME/CFS can be masked by simply indicating fulfillment of case definitions. Other standardized measures of illness domains are needed to determine how these patients may correspond to those in other studies. (fatigue, pain, sleep, cognition, POTS, etc.) In addition, how long had the patients been ill and what medication or medications were they taking.Author Response
Reviewer 1 comments:
The study is clearly described, and the studies are carefully documented. The major problem is that only 6 patients and 4 controls are tested, rally too small to even be considered a cohort. Nonetheless, carefully conducted studies on even small numbers of ME/CFS patients are of interest to the field. In a reply to a question about measurement error and stability of findings, the authors describe this as an exploratory study. That is an excellent description of the situation, and the term should be included in the title, i.e. “Bioenergetic and proteomic profiling of immune cells in ME/CFS patients – An exploratory study”.
We very much agree with the reviewer and has changed the title of the paper as suggested to:
"Bioenergetic and proteomic profiling of immune cells in Myalgic Encephalomyelitis/Chronic Fatigue Syndrome patients – An exploratory study”
The data are encouraging and will be useful to prompt larger studies to explore this question more fully. However, the findings cannot be considered definitive. Abstract uses term “well-defined cohort” (line 25). The patients are well characterized clinically, but well-defined cohort is misleading. The numbers of patients and controls need to be included in the abstract to temper findings.
We agree with the comment and have changed the abstract accordingly (line 26):
“Here, we evaluated patients' symptomatology and the mitochondrial metabolic parameters in peripheral blood mononuclear cells (PBMCs) and plasma from a clinically-well characterized cohort of six ME/CFS patients compared to age- and gender-matched controls.”
The authors state that the study recruited a homogeneous group of ME patients. However little specific information about the patient’s illness, other than indicating case definition fulfilled. The heterogeneity of ME/CFS can be masked by simply indicating fulfillment of case definitions. Other standardized measures of illness domains are needed to determine how these patients may correspond to those in other studies. (fatigue, pain, sleep, cognition, POTS, etc.) In addition, how long had the patients been ill and what medication or medications were they taking.)
It is now stated in the Materials and Methods section, Subsection study design (line 87):
“The patients were asked to abstain from medications – including supplements – a week prior to the examination.”
Regarding duration of illness, it is stated in Table 1 that disease duration in the patients were 12 +/- 5 years. We have included it in the Materials and Methods section, Subsection study design (line 82):
“Six female patients between 30 and 50 years old were enrolled in the study, diagnosed with ME/CFS by a ME/CFS specialist and recruited through the Danish ME-association and with a mean illness duration of 12 years”
Regarding the concern for a more thorough description of the patients: We believe that we have included standardized measures of illness domains as asked by the reviewer by using questionnaires quantifying fatigue (FSMC), autonomic dysfunction (COMPASS-31), and different domains of health-related quality of life including pain (SF-36). Data comparing patients to controls given in Table 2.
Reviewer 2 Report
This is an interesting and comprehensive study of 6 ME/CFS patients compared to 4 controls. The manuscript is overall well written. Care should be taken when drawing conclusion on such small sample, but the main findings agree with previous reports on biochemical alterations in ME/CFS. The authors point to multiple possible elements of the ME/CFS mechanism, which will have to be further validated in subsequent studies.
1) The authors use the term “hypometabolism” several times in the abstract and introduction. This is a mechanistically diffuse term that has not yet been proven to associate with ME/CFS, and my advice would be to avoid the use of such terms since the ME/CFS field desperately needs clarity and testable hypotheses. I even suspect that “hypometabolism” may be misleading and possibly incorrect as a characteristic of ME/CFS. In medicine, hypometabolism reflects a condition marked by an abnormally low metabolic rate (as in hypothyroidism), i.e. a systemic downregulation of metabolism as an endpoint of regulatory shut-down mechanisms. Based on the available data, this does not seem to agree with the findings in ME/CFS patients, but instead there are multiple indications of impaired or stressed metabolism. The authors also conclude that there is mitochondrial dysfunction centered on PDH and CoA metabolism. This may in fact reflect the opposite situation compared to hypometabolism, as the metabolism will need to work harder to fulfill energy demands.
2) Bioenergetics measurements:
These are difficult and sensitive live cell measurements. Traces should preferably be provided to allow transparency and evaluation of the performance. PBMC viability was measured before bioenergetics. What was the viability and how was this implemented? Threshold for inclusion?
The authors address these measurements in their answer to Reviewer 3, and state that the runs were performed on separate days. Information about this, and the experimental control measurements, needs to be detailed in the article. This is not the provided in the current version.
Statistics are not clear: Line 121 “The different parameters were calculated for each well. Then the replicates were pooled for each experimental group.” A mean value of the replicates should be calculated for each subject, which then is used for comparison at group level (ME vs controls).
3) ESM Table2:
- The term “ATP-linked to maximal respiration” is explained incorrectly:
“Oxygen consumed for ATP generation when the mitochondria is uncoupled and protons are moving freely. Theoretical maximal ATP production capacity of the mitochondria without substrate or membrane potential limitations.” In the presence of oligomycin (ATP synthase inhibitor) and FCCP (uncoupler) there is absolutely no ATP generation! A correct description of this calculated parameter would be something like: the fraction of respiratory capacity that is utilized for ATP generation under basal conditions.
- Referring to the classical state 3 and state 4 respiration is not entirely correct and should be removed. These classic bioenergetic states were established using isolated mitochondria and are not identical in the intact cell with membrane barrier and cytosol chemistry.
4) Line 276: “The uncoupling of the mitochondrial respiratory chain may lead to increased leakage of electrons from the electron transport chain, leading to oxidative stress and oxidative damage to biomolecules like proteins”
This statement is unclear and not necessarily true. Uncoupling reflects proton leak, and the impact of uncoupling is context dependent. Uncoupling is in fact also an important physiological antioxidant mechanism (through UCPs). This topic may also be addressed in the discussion.
5) Figure 2: The y-axis title in panels A, C and E should be more specific.
6) Figure 4: It would be more logical to let the arrow point from cytosolic pyruvate to lactate, not from mitochondrial pyruvate.
Author Response
Reviewer 2 comments:
This is an interesting and comprehensive study of 6 ME/CFS patients compared to 4 controls. The manuscript is overall well written. Care should be taken when drawing conclusion on such small sample, but the main findings agree with previous reports on biochemical alterations in ME/CFS. The authors point to multiple possible elements of the ME/CFS mechanism, which will have to be further validated in subsequent studies.
1) The authors use the term “hypometabolism” several times in the abstract and introduction. This is a mechanistically diffuse term that has not yet been proven to associate with ME/CFS, and my advice would be to avoid the use of such terms since the ME/CFS field desperately needs clarity and testable hypotheses. I even suspect that “hypometabolism” may be misleading and possibly incorrect as a characteristic of ME/CFS. In medicine, hypometabolism reflects a condition marked by an abnormally low metabolic rate (as in hypothyroidism), i.e. a systemic downregulation of metabolism as an endpoint of regulatory shut-down mechanisms. Based on the available data, this does not seem to agree with the findings in ME/CFS patients, but instead there are multiple indications of impaired or stressed metabolism. The authors also conclude that there is mitochondrial dysfunction centered on PDH and CoA metabolism. This may in fact reflect the opposite situation compared to hypometabolism, as the metabolism will need to work harder to fulfill energy demands.
We agree with the reviewer, that the metabolism may better be described as “altered” as compared to “hypometabolic” and has changed the wording accordingly – in the abstract and two times in the introduction.
2) Bioenergetics measurements:
These are difficult and sensitive live cell measurements. Traces should preferably be provided to allow transparency and evaluation of the performance. PBMC viability was measured before bioenergetics. What was the viability and how was this implemented? Threshold for inclusion?
We considered a viability threshold of 90%. In our case all the PBMC analyzed had a viability higher than 90%. We have stated it in the materials and methods, subsection Isolation of human PBMCs using Vacutainer Cell Preparation Tube: “The PBMCs viability threshold for inclusion was 90%.”
The authors address these measurements in their answer to Reviewer 3, and state that the runs were performed on separate days. Information about this, and the experimental control measurements, needs to be detailed in the article. This is not the provided in the current version.
We have added following sentence to the methods section:
''To account for inter-run variation, the instrumentation for the bioenergetics measurements was calibrated before every experiment. Moreover, an inter-run blood sample control from a healthy individual with known bioenergetics values, and not related to this study, was drawn on the same day as the study participants and prepared and analyzed at the same time. The measures from the inter-run control sample were not used in the final calculations of bioenergetics values, but used as a control for overall assay performance.”
Statistics are not clear: Line 121 “The different parameters were calculated for each well. Then the replicates were pooled for each experimental group.” A mean value of the replicates should be calculated for each subject, which then is used for comparison at group level (ME vs controls).
In the bioenergetics profile, we referred to experimental group as the three different sequential injections protocols used. We agree with the reviewer that “experimental group” can be confused with the patients and controls. We have rephrased them as the following sentence:
“The different bioenergetic parameters were calculated for each well. Then, the replicates were pooled for each bioenergetic parameter to calculate the mean and the standard deviation.”
3) ESM Table2:
- The term “ATP-linked to maximal respiration” is explained incorrectly:
“Oxygen consumed for ATP generation when the mitochondria is uncoupled and protons are moving freely. Theoretical maximal ATP production capacity of the mitochondria without substrate or membrane potential limitations.” In the presence of oligomycin (ATP synthase inhibitor) and FCCP (uncoupler) there is absolutely no ATP generation! A correct description of this calculated parameter would be something like: the fraction of respiratory capacity that is utilized for ATP generation under basal conditions.
We understand the confusion generated by the definition of the term “ATP-linked to maximal respiration”. It is calculated from the ATP-linked respiration, which is the oxygen consumption associated to ATP generation in basal conditions, and the maximal respiration, which is the theorical maximal capacity of oxygen consumption measured during uncoupling. We have adjusted the definition in the ESM Table 2 to: “Theoretical maximal ATP production capacity of the mitochondria without substrate or membrane potential limitations.”
- Referring to the classical state 3 and state 4 respiration is not entirely correct and should be removed. These classic bioenergetic states were established using isolated mitochondria and are not identical in the intact cell with membrane barrier and cytosol chemistry.
We agree with the reviewer that the classical bioenergetics states are calculated using isolated mitochondria and that those parameters are not identical to the bioenergetic measurements in intact cells. We have removed the sentences referring to the classical bioenergetics states.
4) Line 276: “The uncoupling of the mitochondrial respiratory chain may lead to increased leakage of electrons from the electron transport chain, leading to oxidative stress and oxidative damage to biomolecules like proteins”
This statement is unclear and not necessarily true. Uncoupling reflects proton leak, and the impact of uncoupling is context dependent. Uncoupling is in fact also an important physiological antioxidant mechanism (through UCPs). This topic may also be addressed in the discussion.
We agree with the reviewer that uncoupling can also be an antioxidant mechanism. In both cases, changes in oxidative stress can be observed and we evaluated them by measuring oxidative damaged proteins.
We have revised the sentence:
“The uncoupling of the mitochondrial respiratory chain is a well-known regulator of reactive oxygen species that may cause oxidative stress and oxidative damage to biomolecules like proteins.”
5) Figure 2: The y-axis title in panels A, C and E should be more specific.
The figure 2 A, C and E correspond to ratios between two parameters. Therefore, there is no unit as ratios have no units. This has been clarified in the ESM Table 2, where units were included in the ratios.
The parameters are calculated in the following manner:
Figure 2A: Coupling efficience = ATP-linked respiration / basal respiration
Figure 2C: ATP-linked to maximal respiration = ATP-linked respiration / maximal respiration
Figure 2E: Spare respiratory capacity = maximal respiration / basal respiration
6) Figure 4: It would be more logical to let the arrow point from cytosolic pyruvate to lactate, not from mitochondrial pyruvate.
We agree, and have changed figure 4 and the graphical abstract accordingly.
This manuscript is a resubmission of an earlier submission. The following is a list of the peer review reports and author responses from that submission.
Round 1
Reviewer 1 Report
Excellent and interesting study.
I have no suggestions for any changes.
Reviewer 2 Report
This paper evaluated symptomatology and mitochondrial metabolic parameters in patients with chronic fatigue syndrome (CFS). The report was well written. However, the extreme low sample size (6 patients and 4 controls) prevented the results from supporting any conclusion. The minimal sample size not only limits the study from false negative, which was not the case in this report and argued by the author in the limitation section. It also may generate false-positive results. Further, although table 1 listed demographic data from 4 controls. No information provided if those controls have a sedentary lifestyle or not.
Reviewer 3 Report
The authors present a study of mitochondrial function in peripheral blood mononuclear cells and correlation with symptoms in 6 patients with ME/CFS and 4 controls based on the hypothesis that ME/CFS is related to abnormal energy metabolism. They present an overview of data supporting the hypothesis and show reduced coupling efficiency in PBMCs of patients. Based on proteomic and metabolomic data they conclude dysregulated mitochondrial metabolism centered on PDH and CoA metabolism are associated with ME/CFS. Additional clarification is requested.
- Authors need to correct citations as at least the first 35 in the introduction fail to provide information related to the statements for which they were cited (I gave up trying to match up at that point). This is disconcerting and takes away from the interest of the paper. The citations are really needed to provide additional information on statements made in the introduction as well as throughout.
- Recruitment of the healthy controls is not described. What was the timing of the testing for cases and controls? Were samples from both groups included in the same runs? What is the timing between blood collection and questionnaires? Were any tests repeated on more than one occasion – what is measurement error and stability of findings?
- Figure 1 shows plots for 8 to 9 people. Case and controls should be designated, reasons for omission of points should be provided. As could be expected, the controls appear to be separate from cases and may be the reason for the observed correlation between fatigue and symptoms. The findings should be evaluated among cases to support statement (line 253-254) “physical health of the patients is connected to fatigue and ANS dysfunction.”
- In figure 2A, G, and H only 9 points are shown – reason for omission should be provided. Unless all participants are included for any measures – omission needs to be acknowledged and explained.
- The authors cite similar levels of CD proteins between cases and controls as evidence that distribution of cell types in PBMCs were the same. This is not particularly robust. Why wasn’t a direct analysis of cell types included?
- The study is quite limited, and while this is noted by the authors, the discussion includes many statements as definitive rather than suggestive. Data does not fully confirm a dysregulated mitochondrial metabolism.
- ESM is used without definition – What does it stand for?